# Assessing the quality of information extraction

## Abstract

Advances in large language models have notably enhanced the efficiency of infor-
mation extraction from unstructured and semi-structured data sources. As these
technologies become integral to various applications, establishing an objective
measure for the quality of information extraction becomes imperative. However,
the scarcity of labeled data presents significant challenges to this endeavor. In
this paper, we introduce an automatic framework to assess the quality of the in-
formation extraction/retrieval and its completeness. The framework focuses on
information extraction in the form of entity and its properties. We discuss how to
handle the input/output size limitations of the large language models and analyze
their performance when extracting the information. In particular, we introduce
scores to evaluate the quality of the extraction and provide an extensive discussion
on how to interpret them.

## 1 Introduction

In the domain of natural language processing (NLP), information extraction (IE) stands as a critical
task, transforming unstructured or semi-structured data into a structured format conducive to indexing,
exploration, and further analysis. The increasing amount of data across digital platforms underscores
the urgency for sophisticated IE techniques that can parse through volumes of information with
precision. An extensive survey about IE is provided by [1], where the authors highlight the complexity
of processing and analyzing text to derive meaningful information, given the heterogeneity and volume
of such data.

Large language models (LLMs) have revolutionized IE by introducing generative methods for
structuring knowledge from text. LLMs excel across diverse domains without extensive task-specific
training. A survey by [9] details the progress of LLMs on IE tasks. Here, the authors address specific
aspects of information extraction, including entity recognition, relation extraction, event detection,
and universal IE. They review the existing models and their efficiency on a comprehensive collection
of annotated benchmarks. Nonetheless, the challenge of quantitatively assessing the quality and
completeness of extracted information persists, particularly in the absence of labeled datasets for
benchmarking. Before conducting the experiments introduced in this paper, we perform IE on a vast
corpus of business documents utilizing LLMs. While the extraction process is beyond the scope of
this paper, some details about the extraction are given in Section 3.

To measure the quality of extraction, we propose an evaluation framework that relies on artificially
generated complex information which is infused into the document to test the efficiency of LLMs in
IE tasks. This paper introduces an iterative extraction process and a novel score, MINEA (Multiple
Infused Needle Extraction Accuracy), to address the critical need for objective quality assessment
measures. By inserting artificial information ("needles") into the data, the proposed method creates
a synthetic ground truth for evaluation, enabling the measurement of extraction quality in various
specific domains even without manually labeled data. The empirical analysis demonstrates the
utility of MINEA for evaluating LLM-based IE in scenarios where ground truth is unavailable. By

Submitted to 38th Conference on Neural Information Processing Systems (NeurIPS 2024). Do not distribute.

automating the quality assessment of information extraction, the framework could reduce the need for manual review by experts, saving time and resources and thus enhance the efficiency and accuracy of information extraction from large volumes of unstructured data.

The paper is organized as follows: Section 2 presents a related work that inspired us when developing our IE quality assessment method; Section 3 sketch a way in which structured information is obtained using LLMs; Section 4 deals with shortcomings arising when treating long contexts by LLMs; finally Section 5 introduces the novel method to access the quality of IE and provide the reader with practical tips; Sections 4 and 5 are supplemented by numerical studies. The data used in these studies are an internal set of documents related to a business case in the healthcare industry.

## 2  Related work

A common practice in many specialized IE tasks is that well-trained experts review what was extracted and provide ground truth as done in [5]. Such an approach is relatively reliable, however, it is manual and very time-consuming.

In [4] they suggest *summary score without reference* (SUSWIR), a score to evaluate the quality of text summaries without the need for human annotations. The SUSWIR score can be used for IE tasks where the extracted information is viewed as a compression of original data. The score compares the original text with its summary. From its nature, it is very useful when comparing the outputs of extraction tasks among themselves, i.e., the best extraction/summary has the highest score value. On the other hand, its ability to provide an objective absolute evaluation of a single extraction is disadvantaged because the desirable output is not known.

Recently, an effort to eliminate the requirement for human involvement relies on LLMs. These prove themselves as highly cost-effective data creators, either by labeling unlabeled data or generating data given the labels, see [7]. Therefore they may substitute human experts providing the ground truth by doing their work in an automatic way.

Needle In A Haystack (NIAH)[1] evaluation is a tool designed to evaluate the performance of LLMs in retrieval across different sizes of context. Short targeted information, the 'needle', is inserted into a large, more complex text body, the 'haystack'. The goal is to test an LLM's ability to find and make use of this piece of information.

Our method builds on LLMs acting as data creators, but instead of annotating the complete data, it only automatizes the process of creating the needle. I.e., given an original text, an LLM generates the needle. The needle then substitutes the ground truth.

## 3  Capturing the structure

The form of needles depends on a form of data, on structure capturing the information and on the task being solved. The needles can be short paragraphs of text, account records, graph nodes as you extract information from continuous text, table, graph, respectively. The structured arrangement of information is beneficial for consecutive processing and analysis. It helps to highlight relationships among distinct information pieces. There are countless ways to impose a structure on unstructured data in order to capture the relevant information. To demonstrate our methodology for measuring the quality of information extraction, we specify a particular structure and tailor the needles to it.

### 3.1  Schema

To impose a structure on the data, we adopt the idea of schema markup [3] which is used to communicate the content of a web page to the search tool. The schema markup is in the form of structured data and can be viewed as a compression of the essential information. The structure is defined by Schema.org[2] vocabulary which is a set of entity types, each associated with a set of properties and hierarchically arranged. Figure 1 shows an example of structured information inspired by Schema.org. It describes three entities of types 'Insight', 'Person' and 'Organization'. Each

---

[1]https://github.com/gkamradt/LLMTest_NeedleInAHaystack
[2]https://schema.org

type has its own set of properties, e.g., an entity of type 'Person' is described by 'type', 'name', 'birthDate', 'worksFor', and 'jobTitle'. In other words, each entity is a set of key-value pairs, e.g., 'name' is the key and 'AI Enthusiast' is the value.

```
[
  {
    "@type": "Insight",
    "name": "Information exctraction tested by Needle in a Haystack test",
    "description": "A short targeted information pieces, the 'needles', are inserted to
        a large, more complex text body, the 'haystack'. The quality of information
        extraction task is measured by ratio of succesfully extracted needles.",
    "keywords": "information extraction (9), large language models (8), quality evaluation
        (10), needle in a hayastack (8), named entity recogniction (7), schema.org (6)",
    "author": {
        "@type": "Person",
        "name": "AI Enthusiast",
        "birthDate": " ",
        "worksFor": {
            "@type": "Organization",
            "name": "Creative Dock"
            "description": "Creative Dock builds and scales disruptive tech companies, as
                a startup and corporate venture builder. The company provides end-to-end
                venture-building, from idea to building and scaling.",
            "keywords": "tech company (8), venture builder (9), AI (7), startup growth (8)"
        }
        "jobTitle": "Data Scientist"
    }
  }
]
```

Figure 1: Toy example: structured information encapsulating three entities using schema.org.

Similarly, we extract and compress the relevant information contained in data using an LLM. Schema.org presents a clear basis for the categorization of various entities contained in data. In the rest of the paper, by schema we mean a predetermined set of types, such as {'Person', 'Project', 'Product', 'Legislation', 'Event', 'OpportunityArea', 'Insight', 'Substance', 'Thing', 'BioChemEntity', 'MedicalCondition'}, together with their properties. The schema is set at the beginning and the information to be extracted depends on it. Therefore the schema has to be tailored to a particular scope of the (proprietary) knowledge and application. If a more complex or uncommon entity needs to be captured, it is natural and very easy to extend the set of core types by more detailed descriptive and custom vocabulary. E.g., 'Insight' and 'OpportunityArea' are not native Schema.org types, but we will use them in our study. The usage of suitably tailored schema is beneficial for specialized applications since it narrows the information to the relevant core and hence potentially improves the overall performance. On the other hand, the usage of schemata is not restrictive as the scope can be always extended by using a broader set of types.

## 3.2 The role of LLMs

LLMs are rather effective in the creation of structured data, cf. [9]. Using dedicated prompts, we get a structured text file describing entities found in the documents and matching types of predefined schema. The predefined schema (types and properties) is given to an LLM within the prompt. The LLM is asked to analyse the document, identify an information relevant to the mentioned types of entities and populate the schema with this information. It is asked to be attentive to nested entities, maintain consistency and uniqueness of extracted entities. Indeed, LLM is not prohibited from extracting entities whose types do not appear in the predefined schema. It is worthy to note, that LLMs are known to inherit biases present in their training data. If not carefully managed, these biases could lead to unfair or inaccurate information extraction, impacting decision-making processes.

Besides the information extraction task, LLMs can be used to suggest suitable Schema.org types for a particular document. An example together with a prompt is shown in Appendix B1.

# 4   Length aspects

When focusing on the quality of IE performed by an LLM, several limitations that LLM presents in terms of the length of data to be extracted from must be considered. Each LLM has a maximal content limit it can process, both on the input and the output. The limit on the output is typically much more strict. When trying to use the maximal possible input another issue may appear – the *Lost in the middle* phenomenon [8] says that the ability of LLMs to retrieve information from a long context declines and that the attention focuses on the beginning and the end of the context while it tends to attenuate information in the middle.

To demonstrate shortcomings arising from these limitations numerically we use *gpt-4-1106-preview* model.[3] The model is limited by 4095 tokens on the output and by 128000 tokens on the input (context window limit). The following sections present two major LLM limitations we have to consider before performing IE, namely length restrictions in Section 4.1 and *Lost in the middle* problem in Section 4.2.

## 4.1   Length restrictions

Long data are difficult to process because of the restrictions posed by the maximum amount of:

(O) output tokens: The restriction on output tokens means that there is some maximal length of data from which most entities can be efficiently extracted. If the length of the text exceeds this maximum, there would be no tokens for extra entities.

(I) input tokens: Maximal size of context window (input) prohibits the extraction of data exceeding the specific token limit.

Another difficulty regarding the output is the tendency of LLMs to generate rather brief responses which do not use the allowed maximal number of tokens. This unwillingness of models can be circumvented by prompting. Even so, the limited number of output tokens is typically too low and prevents effective extraction from long texts.

With a more sophisticated approach, the restriction (O) becomes irrelevant and only the restriction (I) will apply. The issue imposed by (O) is overcome by splitting the source document into smaller pieces which are extracted independently. A significant drawback is that the extracted information can be easily duplicated – extracted independently from multiple text pieces. Iterating the calls to the LLM with instruction to continue with already started extraction, i.e., continuing with the extraction in a single thread, helps to extract more information and avoid duplication. As we insist on continuation, more and more information is added and the extraction is more thorough, at least to some point – this will be addressed in detail in Section 5.1. Further, a lower number of duplicates is found due to the extraction history, i.e., all information extracted until present, which is kept within the thread.

The combination of both improvements – text splitting and iterated calls, has proven itself to perform the best. We split the document into distinct text pieces which we extract sequentially. Extraction from each text piece is carried out by several iterated LLM calls while taking into account the extraction history from previously extracted text pieces. Once the sum of the lengths of the text pieces and the extraction history exceeds the context window limit, i.e., restriction (I) applies, a new independent extraction starts. A single structured output, per document or once (I) is applied, is created by appending all entities identified from each text piece.

## 4.2   Lost in the middle

In the case of long documents, whose extraction consumes almost the whole context window, LLMs are giving more inconsistent results and we can observe a presence of the *Lost in the middle* phenomenon, see [8]. We extract information from several long documents from our business case which are each split into 15 pieces and its processing consumes almost the whole context window. We add the sixteenth piece identical to one of the fifteen that are already extracted and measure a *redundancy* score, for details see Appendix A. Each column of Table 1 then states the redundancy of the newly extracted information with the information that was already extracted from the same piece of the text before. The table presents mean values per four distinct documents. We can notice that

---

[3]https://platform.openai.com/docs/models/overview

for the parts 'in the middle' the proportion of redundantly extracted entities (entities with the same 'name' attribute) is higher than for those at the beginning and the end.

Table 1: Are we lost in the middle? After finishing the extraction of a whole document (consisting of fifteen pieces), we re-extract the information from each of its pieces. Columns 1-15 then compare the re-extracted information with the information that was extracted from the same piece of the text before. The pieces in the middle of the document contain more duplicated entities then those at the beginning and the end.

| part | 1 | 2 | 3 | 4 | 5 | 6 |
| --- | --- | --- | --- | --- | --- | --- |
| redundancy (key = 'name') | 0 | 0 | 0.2266 | 0.1150 | 0.1482 | 0.3816 |

| 7 | 8 | 9 | 10 | 11 | 12 | 13 | 14 | 15 |
| --- | --- | --- | --- | --- | --- | --- | --- | --- |
| 0.3334 | 0.4643 | 0.7398 | 0.5152 | 0.6672 | 0.4659 | 0.3820 | 0.4473 | 0.4086 |

# 5    Quality of extraction

Once the information is extracted from data into a structured form defined by the chosen schema, e.g., Figure 1, the quality of such extraction is important to evaluate. In practice, it is very rare to be equipped with ground truth and its human generation requires vast expertise in the scope of data and a ridiculous amount of time. Therefore we adopt methods from [4]. They examine semantic similarity, relevance, redundancy, and bias and compound these into a single score called SUSWIR, for details see Appendix A. The score and its subparts are very useful when comparing distinct extractions among themselves, e.g., we can use it to find an optimal number of iterated LLM calls. Unfortunately, the score does not represent an absolute way of evaluation. It does not provide a complete insight into the task – some information (= entities) can be missing, misclassified or their properties not filled in correctly. To come up with a robust and general solution we generalize the NIAH test, which is commonly used to measure the ability of LLMs to process long documents, cf. [6].

## 5.1    Iterated LLM calls

Since the first LLM extraction is typically not exhaustive, iterating the extraction process helps with the completeness of extraction. To improve the quality of extraction, we ask LLM to process the document again and search for other entities which were not extracted yet. A question arises: What is the optimal number of iterations? It is desirable to stop when additional LLM call will return no or only a few new entities. The answer however depends heavily on the text being extracted and on the chosen schema. Below, we present a small comparative study regarding the contribution of iterated extraction to its quality. We interpret the extracted structured data, e.g., Figure 3, as a summary of the original text document. To measure the quality of the summary we adopt the scores from [4] (a convex combination of these scores creates the overall SUSWIR metric), namely *semantic similarity*, *relevance*, and *redundancy avoidance*. We use a modified *bias avoidance* score from [4] and add two new scores, *relevance spread*, and *incompleteness score*. See Appendix A for more details.

Consider document which length is approximately 12k chars. Table 2 compares the content of the document with extracted information created iteratively by succeeding LLM calls. Each iteration enriches the extracted information, but the benefit decreases. From the third iteration, i.e., after four LLM calls, the majority of scores in Table 2 are either getting worse or stagnating (the arrows following the score name indicate the direction in which the score improves). It is obvious that shorter and longer text will require less or more iterations to extract majority of information without reducing its semantic and factually relevant meaning, respectively. Further, the risk that the LLM will suffer from hallucinations increases as we observe a growth of bias. In the rest of the paper we use three iterations to extract documents of approximate length 12k chars within all extractions (if not stated otherwise).

## 5.2    Test the quality

This section introduces a robust and versatile score to objectively measure the quality of IE. Assuming the structure is imposed by some schema, see Section 3.1, we would like to measure the IE quality as

Table 2: Quality of extraction depends on a number of calls to LLM. The first iterated call is the most beneficial one. From some point (bold) the scores stagnate or even deteriorate. All scores have values between 0 and 1, the arrows indicate whether lower ($\downarrow$) or higher ($\uparrow$) values are desired.

| # iterations | 0 | 1 | 2 | 3 | 4 | 5 |
|---|---|---|---|---|---|---|
| semantic similarity $\uparrow$ | 0.5416 | 0.6316 | 0.6899 | **0.7572** | 0.7540 | 0.7685 |
| relevance $\uparrow$ | 0.3409 | 0.4396 | 0.4449 | **0.4746** | 0.4522 | 0.4445 |
| relevance spread $\downarrow$ | 0.3364 | 0.2493 | 0.2350 | **0.1445** | 0.1428 | 0.1368 |
| redundancy avoidance (0.2) $\uparrow$ | 0.7727 | 0.8670 | 0.8810 | **0.9257** | 0.9251 | 0.9307 |
| redundancy avoidance (0.1) $\uparrow$ | 0.4697 | 0.5936 | 0.6854 | **0.8002** | 0.7972 | 0.8119 |
| redundancy avoidance (0.5, key='name') $\uparrow$ | 0.8182 | 0.9163 | 0.9422 | **0.9650** | 0.9699 | 0.9726 |
| bias avoidance $\uparrow$ | **0.5614** | 0.5515 | 0.4925 | 0.4559 | 0.4447 | 0.4247 |
| incompleteness $\downarrow$ | 0. | 0.5862 | 0.6735 | 0.4217 | 0.5413 | 0.4615 |

a portion of successfully extracted entities, i.e., the accuracy of name entity recognition (NER) task taking into account even the context captured by entity properties. Unfortunately, such an experiment is unfeasible without labeled data. As a consequence, it is unfeasible in many specialized tasks because of the absence of suitable labeled data unseen by LLM models. This can be the case with very recent datasets as well as proprietary datasets. To overcome this issue we use inspiration by NIAH test to build up an automatic and general procedure to access the quality of IE tasks.

### 5.2.1 Needles

A 'needle' in our context represents an entity. It is created according to the chosen schema, i.e., a list of types we want to extract from the document. We use an LLM to generate a short paragraph introducing a new original (not appearing in the document) entity, but still relevant to the scope of the document, for an example see Figure 2, and for more details on generation process see Appendix B2. This artificial paragraph, the needle, is then placed into the document body at random (taking into the account natural units within the text as sentences, paragraphs, etc. if applicable). Moreover, the needle is accompanied with several properties, namely we assign to the needle a name, short description and keywords, see Figure 2. This additional properties are assigned to the needle by the LLM.

### 5.2.2 Multiple infused needle extraction accuracy

To measure the quality of extraction we propose a *multiple infused needle extraction accuracy* (MINEA) score. Its computation combines the approach of NIAH evaluation and NER task. We scatter several needles at random over the text document body (such that the inserted needles fill 10 to 30% of the enriched text) and measure how many of them were successfully extracted. Since we know what exactly was inserted, we know what should be extracted. Then we can objectively measure the quality of extraction on these new entities and moreover, we can compare extracted information from the document with and without needles. Table 3 shows extraction accuracy – MINEA score – total and per schema type – measured on a vast corpus of business documents with predefined schema consisting of types 'BioChemEntity', 'Event', 'Insight', 'Legislation', 'MedicalCondition', 'OpportunityArea', 'Person', 'Product', 'Project', 'Substance' and 'Thing'.

### 5.2.3 Identification of needles

Matching the generated needles with extracted entities imposes a challenge and mostly depends on the formulation of needles. If the needles are too complex or too vague, the straightforward identification changes into a serious problem. For this reason, we equip the needles with additional properties which are then used to compare the needles with extracted entities and to decide whether the needles were extracted successfully or not.

We present several alternative ways how to measure whether the extraction of a needle is successful:

**n** an entity with a name perfectly matching the needle name is found;

**ns** the needle name is found among the extracted information;

```
[
  {
    "@type": "Event",
    "needle": "The AI Clan Meeting on Thursday aims to bring together a diverse team for
        collaboration and knowledge sharing. It is a hybrid event, with team members
        gathering in person at the office while also connecting online via video
        conferencing. The meeting will feature discussions on recent AI projects,
        updates on upcoming initiatives, and collaborative brainstorming sessions.",
    "name": "AI Clan Meeting",
    "description": "The aim of hybrid event AI Clan Meeting happening on Thursday is to
        foster collaboration and engagement among the team. The agenda includes project
        discussions, updates on upcoming initiatives, and brainstorming sessions.
    "keywords": "AI (9), AI projects (9), project updates (7), team collaboration (6),
        knowledge sharing (7), hybrid event (4)",
  },
  {
    "@type": "Product",
    "needle": "Graph Index (GRIX) is a cutting-edge retrieval-augmented generation model
        that is based on a knowledge graph. A graph representation of the knowledge base
        enhances effectiveness and ability to answer complex user queries. It is end-to-
        end solution for question-answering task dealing with the knowledge graph
        construction from and the retrieval of a relevant information from it.",
    "name": "Graph Index",
    "description": "GRIX is an innovative retrieval-augmented generation model based on
        a knowledge graph. A great focus is laid on proper extraction of information from
        data, its composition into the graph and retrieval of a relevant subgraph.",
    "keywords": "retrieval-augmented generation (9), knowledge graph (8), information
        extraction (6), product innovation (7), graph index (8), question-answering (8)"
  }
]
```

Figure 2: Toy example: two needles, highlighted by blue color, accompanied by additional information described by 'name', 'description', and 'keywords'.

Table 3: Quality of extraction – MINEA score – total and per schema type. Entity types are grouped into five classes - 1. three most frequent schema.org types in the documents; 2. med-bio-chem entities, somewhat interchangeable types; 3. best distinguishable types; 4. custom (non Schema.org) types; 5. Schema.org types related to documents, but not stated in the chosen schema. Note: an entity is assumed to be extracted if it is contained within the extracted information - often its type can be misclassified (Project-Product-OpportunityArea, Substance-Thing-BioChemEntity) or sometimes it can be mentioned indirectly (Organization is related to a Person by property 'works for').

| class | entity type | extraction accuracy | # entities used for evaluation |
|---|---|---|---|
| 1 | Person | 0.884 | 69 |
|  | Project | 0.702 | 47 |
|  | Product | 0.750 | 52 |
| 2 | Substance | 0.822 | 45 |
|  | Thing | 0.739 | 46 |
|  | BioChemEntity | 0.674 | 43 |
|  | MedicalCondition | 0.636 | 44 |
| 3 | Legislation | 0.942 | 52 |
|  | Event | 0.915 | 47 |
| 4 | OpportunityArea | 0.671 | 73 |
|  | Insight | 0.747 | 91 |
| 5 | Organization | 0.907 | 43 |
|  | Place | 0.767 | 43 |
|  | overall | 0.780 | 695 |

**k** an entity with some number of keywords perfectly matching the needle keywords is found, the number is determined by the threshold parameter determining the percentage of keywords to be matched;

```
[
  {
    "@type": "Insight",
    "name": "Information exctraction tested by Needle in a Haystack test",
    ...
  },
  {
    "@type": "Event",
    "name": "AI Meeting",
    "description": "A hybrid event bringing together a diverse team for collaboration and
        knowledge sharing.",
    "keywords": "AI Clan Meeting (9), collaboration (8), knowledge sharing (8), hybrid
event (7), team gathering (7), video conferencing (6)"
  },
  {
    "@type": "Product",
    "name": "GRIX",
    "description": "Cutting-edge retrieval-augmented generation model based on a knowledge
        graph",
    "keywords": "GRIX (10), retrieval-augmented generation (9), knowledge graph (10),
        question-answering (8), graph construction (6), information extraction (7)"
  }
]
```

Figure 3: Toy example: extracted information from the data infused by needles from Figure 2.

**llm** an entity matching the needle according to LLM is found.

Table 4: Toy example: fulfillment of the conditions. The text enriched by two needles from Figure 2 was extracted into the form shown in Figure 3.

| entity type | condition for needle identification | | | | | |
| --- | --- | --- | --- | --- | --- | --- |
| | **n** | **ns** | **k0.5** | **k0.6** | **k0.7** | **llm** |
| Event | 0 | 1 | 1 | 0 | 0 | 1 |
| Product | 0 | 0 | 1 | 1 | 0 | 1 |

Note that other conditions can be constructed, e.g., based on the short description instead of keywords, etc. Table 4 shows whether the conditions are fulfilled in the example illustrated by Figures 2 and 3. Namely, the condition **n** is not satisfied ('AI Clan Meeting' $\neq$ 'AI Meeting', 'Graph Index' $\neq$ 'GRIX'). Condition **ns** is satisfied only for needle representing an entity of type 'Event' ('AI Clan Meeting' can be found in the extracted information). There are three keywords out of the six assigned to the needle representing the entity of type 'Event' which match the keywords of an extracted entity, hence **k0.5** is, and **k0.6**, **k0.7** are not satisfied (there is an entity within the extracted information with $50\%$ of keywords being the same as the keywords of the needle). In the case of the second needle, there are four such keywords, therefore **k0.5** and **k0.6** are satisfied. Finally, both needles are identified within the extracted information by an LLM.

Table 5 shows scores (ratios of successfully extracted entities) based on the above criteria in the case of our business documents. The types of inserted needles are 'BioChemEntity', 'Country', 'Event', 'Insight', 'Legislation', 'Person', 'Product', 'Project' and 'Substance'. Matching the needle and entity name usually does not perform well if the name is prone to modification (e.g., person name with and without title), or if the entity is easy to be misclassified (an entity of type 'Country' was often extracted as 'Place' whose name did not match the country name). Searching for a needle name in all extracted information gives very accurate results if the entities are well characterized by their name (compare for example types 'Person' and 'Legislation' with type 'Insight' where the name is not a natural attribute). Matching the needle and entity keywords depends on the threshold parameter – with a lower proportion of keywords that have to match the score value increases and the reliability of the entity identification decreases. An LLM performs well the entity identification and it is an important criterion in the case of more creative types such as 'Insight'. Finally, the MINEA score for each type is taken as the maximum of the scores (the values are highlighted).

Table 5: The decision about the success of needle extraction can be made based on several criteria: comparing the corresponding needle and entity properties (columns **n** and **k0.5**-**k0.7** compare name and keywords, respectively), full-text search (column **ns** search for the needle name in extracted information), comparison of needles and entities using LLM (column **llm**).

| entity type | condition for needle identification | | | | | | # entities used for evaluation |
| | **n** | **ns** | **k0.5** | **k0.6** | **k0.7** | **llm** | |
| --- | --- | --- | --- | --- | --- | --- | --- |
| Person | 0.594 | **0.884** | 0.652 | 0.362 | 0.232 | 0.826 | 69 |
| Project | 0.170 | **0.702** | 0.638 | 0.234 | 0.085 | 0.681 | 47 |
| Product | 0.596 | 0.712 | 0.462 | 0.192 | 0.135 | **0.750** | 52 |
| Country | 0 | **0.765** | 0.412 | 0.294 | 0.059 | 0.471 | 17 |
| Legislation | 0.635 | **0.942** | 0.365 | 0.269 | 0.096 | **0.942** | 52 |
| Event | 0.830 | 0.851 | 0.638 | 0.511 | 0.149 | **0.915** | 47 |
| Insight | 0.176 | 0.187 | 0.714 | 0.418 | 0.088 | **0.747** | 91 |
| BioChemEntity | 0.116 | 0.605 | 0.651 | 0.581 | 0.488 | **0.674** | 43 |
| Substance | 0.289 | 0.578 | **0.822** | 0.644 | 0.222 | 0.800 | 45 |

### 5.2.4  Model comparison

MINEA score can be used to compare the performance of distinct LLMs, see Table 6. A corpus of documents is infused by needles representing entities whose types match the schema introduced in Section 5.2.2. Three OpenAI LLMs[4] are used to extract a relevant information under the same setting (the same model parameters such as temperature, the same number of iterations, the same prompting, etc.). Model *gpt-3.5-turbo* is outperformed by *gpt-4-turbo* by almost 15% and *gpt-4-turbo* is outperformed by *gpt-4o* model by another 12%. Note that the achieved accuracy is lower than presented in Table 3, since only one iteration instead of three was performed in order to reduce the computational time.

Table 6: LLMs comparison using MINEA score.

| model | gpt-3.5-turbo | gpt-4-turbo | gpt-4o |
| --- | --- | --- | --- |
| MINEA | 0.449198 | 0.593583 | 0.716578 |

## Conclusions

In this paper, we focused on quality evaluation of information extraction (IE) performed by large language models (LLMs). First, we delved into the technical limitations of LLMs complicating the extraction of information from a long context. To extract reasonable information from data it is needed to take into the account features such as context window limits, iterated extractions, extraction history recording and *Lost in the middle* phenomenon. Once the extraction is performed, assessing its quality is essential. However in many customized tasks, a truly objective method is missing, because of the lack of labeled data fitting the scope of the application. The versatile method presented in this paper overcomes the issue by adjustment of the data by insertion of an artificial information, a needle, into it. The artificial information created to this purpose is application and data-specific, but the method itself is applicable generally across the field of IE. By controlling the generation process of the needles, we created a synthetic ground truth that enables us to absolutely measure the extraction quality even when no labeled data is available. We introduced a MINEA score to measure the quality of extraction. The key part is a decision rule on whether a needle was successfully extracted or not. MINEA possibly combines several decision rules into one final score. Our empirical analysis of the MINEA score on a specialized dataset demonstrated its utility for evaluation of LLM-based IE tasks when ground truth is unavailable.

---

[4]https://platform.openai.com/docs/models

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

# Appendix A

To measure the quality of the summary we adopt the methods from [4]: *semantic similarity* combines latent semantic similarity and cosine similarity; *relevance* is measured using METEOR score, see [2], without chunk penalty; *redundancy avoidance* compares extracted entities among themselves using a threshold parameter – entities with a higher cosine similarity are assumed to be redundant; redundancy avoidance can be focused on a single particular property of entities (we use 'name' as this pivotal property).

We modify the *bias avoidance* score from [4] to be $J^*(A, B) = \frac{|A \cap B|}{|B|}$, where $A$ represents the entities in the original text document and we normalize by a number of entities that were extracted, $|B|$. The score controls how much information in the structured file is not present in the original text, i.e., a potential hallucination of an LLM.

We add two new scores: the *relevance spread* is the standard deviation of relevance over the text pieces to which the document is split and normalized by the mean value, its higher values indicate that the extraction from distinct text pieces is unbalanced; the *incompleteness score* just measures the proportion of entities with incomplete information (at least one property value missing or unfilled), e.g., the entity 'AI Enthusiast' in Figure 1 has an unknown 'birthDate'.

# Appendix B

Except for the IE task, LLMs are used in several subtasks within the paper, namely to determine schema types appearing in the document, to create a suitable needles fitting contextually to the

document and to identify whether a needle was extracted or not. In the following, we provide the reader with prompts and examples of these subtasks.

**B1 Discovering a schema**

Figure 4 shows a prompt to obtain the Schema.org types from the attached text – Wikipedia article about IE.[5] An LLM is asked to assign relevance to the types to distinguish the most important ones.

Figure 5 shows the entity types that were deduced from the text, together with their relevance and reasoning for why they were chosen. The most relevant types are those directly mentioned – 'Article', as the webpage content itself is represented as an article, 'SoftwareApplication', and 'WebSite' (all with maximal relevance). The least relevant identified types are generic – 'Thing', as a parent type of many directly mentioned types, and 'LearningResource', as a categorization of the article style.

Figure 4: Prompt to determine a possible suitable schema from a given text – Wikipedia article about IE.

Figure 5: Schema.org types found by an LLM within Wikipedia article about IE.

---

[5]https://en.wikipedia.org/wiki/Information_extraction

## B2 Creating needles

A needle, i.e., a text paragraph fitting thematically to the document, but being new and unique to it, is generated by an LLM using the prompt in Figure 6. The prompt specifies the type of entity that the needle should represent. Multiple needles of the same type can be obtained easily within a single LLM call.

Figure 7 shows ten needles representing the entities of type 'Person' generated based on a Wikipedia article about IE. In the next step properties such as a name, description and keywords can be generated by an LLM.

**SYSTEM**

You have been created to test attentiveness and mindfulness of proof readers.

Your task is to suggest a piece of information to be inserted to a document they will carefully examine. The information needs to be completely NEW and UNIQUE, not appearing in the document, but FIT to it THEMATICALLY. A proof reader will pass the test only, if he manage to discover the newly added text.

The proof readers are tested repeatedly. Each test focuses on a particular entity type. The type you are asked to create a piece of text about is 'Person'. Stick only to this type!!! You will be severely penalized for each created entity classified to a wrong schema.org type.

Suggest 10 self sufficient information pieces to be inserted. Use DIFFERENT format and style for each piece in order to confuse the proof readers. Avoid patterns! E.g., placing the name of the entity always at the beginning is a typical pattern behavior. DO NOT HIGHLIGHT the name of the entity.

*Type: Person

*Output: texts to insert

*Document to insert to:
https://en.wikipedia.org/wiki/Information_extraction

Figure 6: Prompt to generate needles. Given a Wikipedia article about IE, the LLM is asked to think out 10 relevant persons.

ASSISTANT

1. Arnold Jennings is a renowned expert in the field of Information Extraction and has published several groundbreaking research papers on the topic in prestigious journals.

2. In 1995, Professor Elena Martinez pioneered a new approach to Named Entity Recognition, revolutionizing the way entities are identified in natural language processing.

3. Dr. James Carter, a leading figure in the field of Information Extraction, will be delivering a keynote speech at the upcoming International Conference on NLP and Machine Learning.

4. Vanessa Rodriguez, a rising star in the world of Information Extraction, recently developed a cutting-edge algorithm that significantly enhances the accuracy of event extraction from texts.

5. The collaboration between Dr. Michael Nguyen and Dr. Sophia Lee resulted in the creation of a powerful tool for Relationship Extraction, which has been widely acclaimed in the NLP community.

6. Professor Samuel Brown received the prestigious Excellence in Information Extraction Award for his groundbreaking research on Coreference Resolution in complex textual data.

7. In her latest research project, Dr. Lily Martinez successfully applied Information Extraction techniques to analyze and extract valuable insights from a large dataset of social media posts.

8. The innovative work of Dr. Oliver Walker in Template Filling has paved the way for more efficient extraction of structured information from unstructured documents.

9. Christina Lee, a leading researcher in the field of Named Entity Recognition, has developed a state-of-the-art system that can accurately detect and classify various types of entities in textual data.

10. Professor Nathan Turner's contributions to Semi-structured Information Extraction have been instrumental in advancing the capabilities of automated systems in processing and interpreting diverse types of data sources.

Figure 7: Needles generated by an LLM and representing ten entities of type 'Person'.

354

### B3 Identifying needles

The quality of extraction is evaluated based on the proportion of successfully extracted needles. An LLM can be used to decide whether the needle was extracted or not using the prompt presented in Figure 8.

**SYSTEM**

You are an expert in named entity recognition (NER).

Given a list of entities your goal is to decide for each of them whether in the attached structured list is an element directly matching to it (the element and the entity have to be one to one correspondence). Each entity has a type, name, description and keywords. Take care about these attributes when making the decision.

For EACH entity in the list you are supposed state a simple YES or NO and in the case of positive answer output the matching element, in the case of negative answer output by meaning the closest element.

Rule: number of decisions = number of entities

You will be strictly penalized for doing decisions upon entities which are not mentioned!!!

List of entities: {list_ents}

Attached structured list: {extracted}

Figure 8:  Prompt to identify whether the needles were extracted or not.

