# OpenReview forum: "Assessing the quality of information extraction"
_NeurIPS.cc/2024/Conference — Submitted to NeurIPS 2024_

### Official Review · Reviewer_W8Fk · 2024-07-07

**Soundness:** 1
**Presentation:** 1
**Contribution:** 1
**Rating:** 3
**Confidence:** 4

**Summary:**

* This paper studies the evaluation of information extraction, particularly LLM-based IE, in scenarios where human-annotated data is unavailable.
* The proposed evaluation framework relies on the `Needle in a haystack` evaluation. That is, an LLM is first used to generate a piece of information (needle) given the original text; then, the needle is infused into the document, and the quality of IE is assessed by whether the needle can be successfully extracted.
* In addition to this evaluation framework, the authors also discussed several aspects to be considered when using LLM-based IE for processing long documents.

**Strengths:**

An interesting application of `Needle in a haystack` evaluation in information extraction.

**Weaknesses:**

* The writing quality is not great, and several areas require further clarification
	* The current paper structure is confusing; not sure what role Sections 3 and 4 play in this paper, e.g., whether the authors were proposing a new LLM-based IE approach
	* I suggest providing a formal definition of IE studied in this paper because it is very confusing to know what information is extracted. For example, in the abstract, `entity and its properties` is mentioned; in Section 3, `short paragraphs of text` seem to be the information extracted `from the continuous text`; also see Q2
* The main contribution of the paper is an automatic framework to assess the quality of the IE; however, the authors didn't conduct any experiments to demonstrate the effectiveness of the proposed framework (e.g., whether the evaluation results correlate with human judgments); the other main limitation is the authors evaluate the quality of extraction based on the proportion of successfully extracted needles but totally ignore the correctness of extracted information (precision)
* The experiments are conducted on private datasets with only several toy examples described in the paper; it will be very difficult for others to reproduce the results. I would suggest conducting experiments at least on some document IE datasets, for example, from news or biomedical domains.

**Questions:**

1. Line 7: Information retrieval is mentioned once but not explained anywhere else; suggest clarifying its meaning in this paper
2. Figure 1: should the value of these fields (e.g., name, description, keywords) be directly copied from the original text? what do these numbers (9) (8) in the `keywords` field mean?
3. Table 1: what higher redundancy scores (more duplicated entities) mean? How do these results tell the `Lost in the Middle` phenomenon?
4. What LLMs do you use to generate the needles and for identifying needles? How does this affect the extraction models to be tested? e.g., is the model more likely to achieve better performance if the needle is created by the same model? does the model achieve better scores if the same model is used for evaluating whether the needle is found (`llm` column in Table 4)?
5. Table 3: what does `chosen schema` mean? Do you mean the LLM is not instructed to recognize entities belonging to these categories, but they are still recognized?
6. What evidence (empirical results) can support the claim that 'the combination of both improvements --- text splitting and iterated calls, has proven itself to perform the best (line 146)'?

---

### Official Review · Reviewer_bpwS · 2024-07-07

**Soundness:** 1
**Presentation:** 2
**Contribution:** 2
**Rating:** 3
**Confidence:** 4

**Summary:**

This paper focuses on the quality evaluation of information extraction (IE) performed by large language models (LLMs). It discusses the methods to handle the input/output size limitations of the LLMs and their performance in IE. It also introduces additional scores to evaluate the extraction quality and discusses how to interpret them.

**Strengths:**

1. This paper analyses the technical limitations of LLMs complicating the extraction of information from a long context.
2. This paper presents to insert a needle into the data to evaluate the performance of IE without labeled data.

**Weaknesses:**

1. The analysis of the performance of LLMs in IE is not new and has various analysis, such as in the following papers:

> [1] Evaluating ChatGPT's Information Extraction Capabilities: An Assessment of Performance, Explainability, Calibration, and Faithfulness (Li et al., 2023)

> [2] Is Information Extraction Solved by ChatGPT? An Analysis of Performance, Evaluation Criteria, Robustness and Errors (Han et al., 2023)

> [3] When does In-context Learning Fall Short and Why? A Study on Specification-Heavy Tasks (Peng et al., 2023)

Among the papers, the authors in [3] also analysed LLMs' limitations in long context understanding, which is similar to the conclusion of this paper.

2. This paper lacks a thorough literature review in LLM for IE as well as new evaluation formats, such as [1, 2, 3] and the following paper:

> [4] Evaluating Generative Language Models in Information Extraction as Subjective Question Correction (Fan et al., LREC-COLING 2024)

3. This paper only focuses on the NER task but lacks the other IE tasks, e.g. relation extraction and event extraction. Additional experiments are required to test the generalisability of the method. The number of samples tested is also limited (see "# entities used for evaluation" in Table 3).

**Questions:**

See "Weaknesses".

**Limitations:**

See "Weaknesses".

---

### Official Review · Reviewer_1gKP · 2024-07-09

**Soundness:** 1
**Presentation:** 1
**Contribution:** 1
**Rating:** 1
**Confidence:** 5

**Summary:**

The paper introduces a framework to capture information extraction quality in the absence of humanly labelled and curated datasets. It explains how an approach on how to include the schema, and the role and limitations of LLM's (specifically gpt-4-1106-preview).

Experiments are done (I guess), by "extracting information" from long business documents originating from the healthcare sector. Several scores are presented according to the SUSWIR metrics. It delves into the "lost in the middle" phenomenon. It introduces the MINEA score, a newly proposed metric.

**Strengths:**

It tries to address a relevant problem in the field (curated benchmark data is hard to come by).

**Weaknesses:**

- The paper is from the start extremely vague and misses concrete statements and explanations about the work done. The contributions are unclear, the data is essentially undefined, for most of the work what exactly is being done is simply unclear.

- Even the task of "Information Extraction" is not concretely described in a way that is reproducible.

- Line 7-8: "The framework focuses on information extraction in the form of entity and its
	properties".

- Table 1: it is completely lost upon me what is being presented here.

-  "We extract information from several long documents from our business case". What are these documents? What are they originating from?

-  The scores mentioned are "redundancy". How is this measured? What do these scores represent? Is lower or higher better? Even these basic questions are not answered. All of this in the appendix (where it shouldnt be), and the further tables are not better.

- The work is very dry. There are no figures that explain or examplify what the problem is, or how this framework is supposed to fit.

- The related work section is short and doesn't address the original point (evaluation in absense of benchmark data).

- It is unclear to me how this work should contribute in any form to evaluation in the absence of benchmark data.

- The introduced MINEA score is "explained", but not examplified or mathematically defined.

- All examples are screenshots of data in JSON format rather than helpful explanations.

**Questions:**

- What do you see as "Information Extraction" in this work?

- What are the concrete contributions of this work

- What is MINEA? And how can it be helpful towards assessing IE?

**Limitations:**

No. The paper does not concretely address the limitations of this metric. There are no good, bad examples provided.

---

### Official Review · Reviewer_CRaf · 2024-07-12

**Soundness:** 2
**Presentation:** 2
**Contribution:** 2
**Rating:** 3
**Confidence:** 4

**Summary:**

The paper proposes an automated framework for evaluating the quality of IE tasks using LLMs. The framework introduces a scoring method called MINEA, which creates evaluation criteria by injecting artificial data ("needles") into documents. The paper also discusses how to deal with the limitations of LLMs when processing large amounts of data, and introduces an iterative extraction process to improve the completeness of the extraction and reduce repetition.

**Strengths:**

s1. The introduction of the MINEA score is somewhat innovative.

s2. The paper is clear explanations of the proposed framework.

**Weaknesses:**

w1. Lack of Originality: The originality of the paper is insufficient. Related work has already mentioned using the "needle" method to evaluate the information extraction capabilities of LLMs. While this paper adds the use of large models to help create the needles, the contribution is still lacking.

w2. Insufficient Experimental Description: The description of the experimental setup is missing, including the experimental environment, data sources, and dataset sizes. However, the paper spends too much space on toy examples.

w3. Unreliable Conclusions on Length Limitations: For the experiments on the input and output length limitations of models, the paper only tested one model, making the conclusions unreliable.

**Questions:**

q1. Please clarify the advantages of this method as the paper does not explain them clearly.

q2. Please provide more details on the specific iterative process and its implementation.

q3. Could you elaborate on the experimental setup, including whether the experiments were conducted multiple times and the reliability of the results?

**Limitations:**

L1. The paper should provide a comparison to existing work to highlight the improvements.

---

### Decision · Program_Chairs · 2024-09-25

**Decision:**

Reject

**Comment:**

This work proposes  an automated framework for evaluating the quality of information extraction tasks using LLMs.
There problem is important but there are key concerns regarding clarity,  proposed solution and evaluation.